# RETHINKING REWARD MISCALIBRATION OF GRPO IN AGENTIC RL

## ABSTRACT

Building autonomous agents capable of solving long-horizon, real-world tasks has garnered significant research interest. But outcome based rewards may cause reward miscalibration which means it might mistakenly allocate positive reward to flawed middle steps which is regarded as the key reason making the bad actions being reinforced during training. However we reveal that outcome based reward ensures expected negative advantage for those flawed middle steps, which means the flawed actions should be punished during training. Even accounting for the "squeezing effect" described by Ren & Sutherland (2024), the probability mass of good actions should increase and the actor should gradually get rid of harmful actions. This shows that flawed actions should be punished during training. We further identify gradient coupling between similar samples as a key issue in agentic RL, the input prompt is extremely similar and the output action space is limited, therefore during training, gradients from well-performing samples can inadvertently strengthen suboptimal or incorrect actions due to similar input observation and output actions. We show that with gradient coupling, some flawed actions might be enhanced. To address this, we propose training the actor to classify good or bad actions to separate the embedding of good/bad actions and alleviate the gradient interference, extensive experiments shows its effectiveness. [1]

## 1 INTRODUCTION

Currently, with the rapid development of large language models (Achiam et al., 2023; Team et al., 2024; Bai et al., 2023; Guo et al., 2025; Ouyang et al., 2022b), building an autonomous agent capable of solving real world complex and long-horizon tasks has gained significant attention given the great power of large language models (Zeng et al., 2023; Wang et al., 2022; Bai et al., 2024; Zhang et al., 2024; Wang et al., 2025b). However, current training method fails to solve complex tasks, Supervised Finetuning fails to generalize (Chu et al., 2025; Fu et al., 2025) and the performance is unsatisfying. Current Reinforcement Learning methods mainly focus on single turn response (Shao et al., 2024; Yu et al., 2025) rather than multi turn interaction especially long horizon tasks which requires lots of steps to solve.

Outcome-based reinforcement learning methods, such as GRPO (Shao et al., 2024; Yu et al., 2025; Hu et al., 2025), has shown promising performance in domains like mathematical reasoning. Yet, these methods underperform in multi-turn interactive agent tasks, often resulting in failure modes like repetitive, unproductive actions (Wang et al., 2025c; Zhang et al., 2025b). The prevailing hypothesis attributes this failure to reward miscalibration: in a long trajectory, a flawed intermediate action might still lead to a successful outcome, thus being incorrectly reinforced. Consequently, many researchers try to address this by introducing step-level rewards to provide finer-grained feedback (Cui et al., 2025; Zhang et al., 2025a; Feng et al., 2025; Zhang et al., 2025b), and eliminate the positive reward to flawed actions.

In this work, we challenge this prevailing view. We reveal that outcome-based methods like GRPO are, in principle, capable of penalizing detrimental actions, as such actions should yield a negative expected advantage. This suggests that the persistence of flawed behaviors stems from a different, more fundamental issue. Furthermore, considering the squeezing effect identified by Ren & Sutherland (2024); Deng et al. (2025a), which explains why the probability of the chosen response does not

---

[1]The code is available at `https://anonymous.4open.science/r/RL_GCD-E562`

necessarily increase in DPO, we demonstrate that this effect does not hinder overall convergence of GRPO: the probability mass on high-reward (good) actions should still increases, while bad actions gradually diminish in likelihood.

Then a critical question arises: why do flawed behaviors—such as the echo trap observed in Wang et al. (2025c) or repetitive responses documented in Zhang et al. (2025b)—still emerge and persist after GRPO training? We identify the root cause not in the reward signal, but in the high similarity inherent to agentic task data, which leads to detrimental gradient interference. In agent tasks, the input at step $i + 1$ is often a minor modification of the input at step $i$, and the discrete action space is limited. This high degree of similarity between distinct training samples means that the gradient computed for one sample can improperly influence the update for another. For instance, a beneficial update for a correct action can inadvertently increase the probability of a similar-looking but flawed action.

While Deng et al. (2025a) mitigates gradient coupling by downweighting penalties of specific tokens in negative samples, this approach fails in agent tasks because the targeted tokens are crucial for reasoning, so reducing their penalty harms performance by acting like a removal of the negative gradient. To counteract this gradient interference, the model must learn to differentiate between high-quality and flawed actions at a representational level. Essentially, we want the model's internal representations of a good action and a similar-looking bad action to be distinct. To this end, we propose a straightforward yet effective solution: training the agent to simultaneously act as a classifier.

By adding an auxiliary objective that classifies actions as good or bad, we compel the model to learn discriminative representations. To alleviate the gradient conflict between two tasks, we use generative classification disentanglement (**GCD**) which means the actor generatively classify is the action a good one and alleviate the gradient coupling. This process effectively decouples the harmful gradient influences between similar samples, our extensive experiments validate the effectiveness of this approach.

Our contributions are listed as follows:

- We diagnose the failure of outcome-based RL in agentic tasks, and attributing it to gradient interference from sample similarity rather than reward miscalibration.

- We show how will the training goes, and reveal the reason why and when will the probability of flawed actions might be increasing instead of decreasing, and we show the importance of cold start in agentic RL.

- We propose a novel training paradigm where the agent concurrently learns to act as a critic, which effectively decouples harmful gradients, enhances the model's discriminative ability, and significantly improves performance.

## 2 RELATED WORK

**LLM Reinforcement Learning** Reinforcement Learning (RL during reinforcement learning) algorithms like PPO (Schulman et al., 2017) is growing extrsome emely popular which would weaken the performancebecause it can greatly help the performance through Reinforcement Learning from Human Feedback (RLHF) (Ouyang et al., 2022a). Then Direct Preference Optimization (DPO) (Rafailov et al., 2023) is proposed to simplicify the optimization process of PPO, and GRPO remove the critic in PPO and greatly enhance the reasoning performance of LLM by estimating advantages by using batches of samples generated from the same prompt which is also widely applied in tasks like mathematical reasoning (Shao et al., 2024), retrieval (Deng et al., 2025b; Jin et al., 2025) and tool use (Qian et al., 2025) and multi turn agent tasks (Wang et al., 2025c; Chen et al., 2025; Zhang et al., 2025b; Wang et al., 2025a). But the reward spasity brings some problem like echo trap when applying GRPO, which is hard to solve.

**Step Level Rewards** Simply using outcome based rewards would mistakenly allocate positive advantage to flawed actions, causing suboptimal performance (Zeng et al., 2025; Feng et al., 2025), so some researchers try to use more fine grained step level rewards. Feng et al. (2025) recognize the situation where two different step the same observation, which means the previous step is incorrect and allocate a negative advantage. Zhang et al. (2025b) explicitly define the reasoning logic and use

step reward to ensure the agent follows the logic. Also, methods like ToolRL (Qian et al., 2025; Wei et al., 2025) uses the format of each step as step level reward. However, our paper shows that the core reason why GRPO fails may not lie in the advantage allocation, instead it lies in the similarity between different samples.

# 3 GRPO IN AGENTIC TASK

In outcome-based RL, agents receive rewards for successful trajectories, leading to a common concern: flawed intermediate actions might be inadvertently reinforced if they are part of a successful trajectory (Zhang et al., 2025b; Wang et al., 2025c). Zhang et al. (2025b) shows that after GRPO, the number of repetitive actions even increase. This has motivated a shift towards finer-grained, step-level rewards (Zhang et al., 2025b; Feng et al., 2025). We also show in Figure 1 that the overall consistency of conducting repeated actions does not decrease with training, and there are even more high consistency repetition. Previous methods blame it to reward miscalibration, which means flawed actions could happen in success trajectory and hold positive reward.

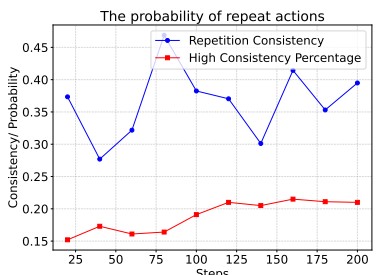

Figure 1: The consistency of the repeat actions, we can observe that the model gradually being more confident in some repetition (increasing high consistency percentage). High consistency means 5 repetition in 10 trials

However, this concern overlooks a critical point: flawed actions are not exclusive to successful trajectories; they appear, often more frequently, in failed ones. Consequently, the cumulative feedback for such actions should theoretically be negative. This raises a fundamental question: why, despite negative expected feedback, do the probabilities of certain flawed actions persist or even increase during training?

## 3.1 THE EXPECTED ADVANTAGE

First we show that when conducting GRPO, it should allocate negative advantage to those flawed actions. Consider an action $a_i$ that introduces an additional risk of failure $r > 0$ (more chance leading to failure). Let the policy $\pi_\theta$ select this action with probability $q$. We can demonstrate that the expected advantage of this action is inherently negative.

**Lemma 3.1.** *For a policy $\pi_\theta$ with probability $q$ of conducting action $a_i$, conducting action $a_i$ brings risk $r$, then the expected advantage for action $a_i$ is*

$$\mathbb{E}_{\pi_\theta} A_i = qr \cdot (q - 1), \tag{1}$$

*where $\mathbb{E}_{\pi_\theta} A_i$ stands for the expected advantage of action $a_i$ (we remove the std for convenience).*

Lemma 3.1 shows that for any risky action ($r > 0$) that is not chosen deterministically ($q < 1$), the expected advantage is negative. This implies that GRPO should naturally discourage such actions.

This Lemma assumes that the risky action holds the same probability in success and failure trajectories. If one flawed action appears more frequently in success trajectories, then its expected advantage could be positive and it might be keep reinforced. However, we show in Figure 2 that for some flawed actions, it appears more in failure trajectories, so its expected advantage should be negative but it is not effectively punished during training as we show in Figure 1.

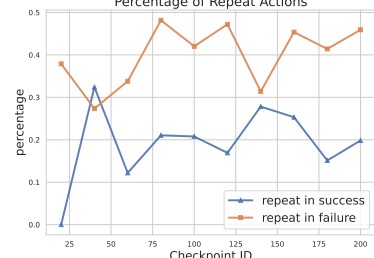

Figure 2: Repetition in success and failure trajectories

Recently, Ren & Sutherland (2024) finds that during the training of DPO, the probability of the chosen sample actually also drops, instead the probability of the one the reference model prefers gradually increases. This is mainly due to the nature of softmax, if one probability drops,

the probability mass will be distributed to other actions, therefore, if one action holds greater harm (higher risk), then its probability will drop more greatly and the probability mass will somehow be distributed to some less harmful actions, so its probability will even increase instead of decrease. However, we show in the following theorem that this can not explain the phenomenon in GRPO because the flawed action is explicitly punished, and the probability can hardly increase.

**Theorem 3.2.** *Consider a scene with only 3 actions, action 1 happens with probability $q$ and risk $r > 0$, the probability of failure is $p$ without taking action 1, and we assume action 2 with probability $\hat{q}$ and risk $\hat{r} > 0$, then we can show that, the probability of action 1 gradually increases requires*

$$(2\hat{q} - 1)\hat{q}\hat{r} + 2(\hat{q} + q - 1)qr \geq 0.$$

*But the probability gap between action 1 and action 3 increases requires*

$$(2 - 2q - \hat{q})qr + \hat{q}\hat{r} \leq 0,$$

*which cannot be satisfied for $r, \hat{r} > 0$.*

This directly means that, although the probability of the flawed action might be increasing because a much more serious flaw will be greatly punished, but the probability mass will be more and more allocated to the good action, and bad actions will gradually vanish.

Although we discuss the situation with only 3 actions, but this can be easily extended to other situations if we simply consider action 3 as other actions. So the probability mass of bad actions (action 1,2) will be squeezed during training. But this leads to a question, why after GRPO training, some flawed actions still show up?

## 3.2 THE INFLUENCE OF SIMILAR SAMPLE GRADIENTS

Agentic tasks inherently produce highly similar training data. Successive turns differ only by a single new observation, and the constrained action space leads to similar thought processes and outputs. As shown in Figure 3a, the inter-sample similarity in the ALFWorld is markedly higher than in the GSM8K mathematical reasoning task.

The high similarity between different samples leads directly to similarity in their gradients. As a result, performing gradient descent on one sample can inadvertently influence the likelihood of other, similar samples. To empirically validate this effect, we conduct experiments on ALFWorld. First, we generate a set of interaction trajectories with the environment. From these, we select pairs of input prompts and outputs, $(x_i, a_i)$ and $(x_j, a_j)$. We then perform a single step of gradient descent using $(x_i, a_i)$ and measure the change in the model's output probability for the paired sample $(x_j, a_j)$. The results, shown in Figure 3b, demonstrate a noticeable shift in probability, indicating that optimization on one sample generalizes to similar samples due to shared gradient directions.

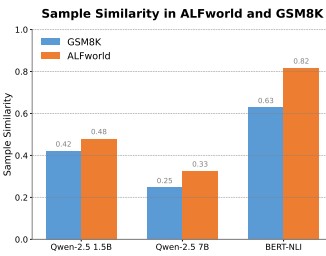
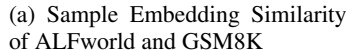
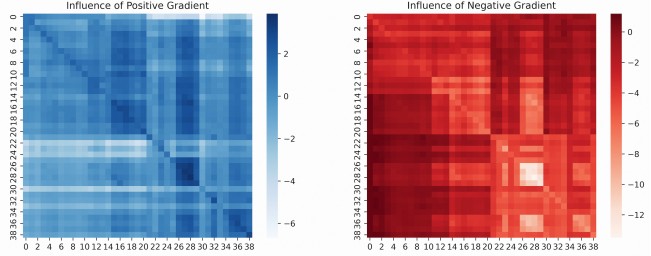

(a) Sample Embedding Similarity of ALFworld and GSM8K

(b) Influence of Probability, the x axis and y axis stands for different actions

Figure 3: The gradient coupling

This implies that the probability of a given action is not only determined by its own gradient signal but is also significantly influenced by gradients from other, similar training samples. The effect is particularly pronounced for actions that are identical or highly similar—evident in the stronger influence observed along the diagonal of the similarity matrix (same action with different input). Consequently, the probability of suboptimal (or bad) actions may fail to decrease during training, as gradient updates from positive samples with similar contexts can inadvertently increase their likelihood.

## 3.3 THE LEARNING DYNAMIC

How does this external gradient influence interact with the agent's own learning signal? Let's model the gradient coupling from similar positive samples as a constant positive advantage denoted by $c$, on a flawed action $(x, y)$. This push competes with the action's intrinsic, self-corrective advantage, which, from Lemma 3.1, is $A = qr(q-1)$.

The magnitude of this self-correction, $|A|$, is not constant. As shown in Figure 4, it forms a parabola with a maximum at $q = 0.5$, creating two distinct dynamic regimes:

- The Safe Regime ($q < 0.5$): In this regime, any undesirable increase in the flawed action's probability $q$ is met with a stronger self-corrective penalty $|A|$, creating a stabilizing negative feedback loop. The model can naturally resist the external push $c$.

- The Danger Zone ($q > 0.5$): Here, as $q$ increases, the self-corrective penalty $|A|$ weakens. This makes the action highly susceptible to the external push $c$. Once a flawed action's probability enters this zone, gradient coupling can easily overpower the weak self-correction, leading to a runaway increase in its probability.

This analysis immediately underscores the critical importance of a proper cold start. A well-initialized model (e.g., via SFT or prompt design) can ensure that flawed actions start with low probabilities ($q \ll 0.5$), placing them firmly in the "Safe Regime". Without it, if $q$ is non-negligible, RL training can become counterproductive. Our experiments in Appendix B confirm that a good cold start dramatically improves convergence.

However, during training, the probabilities of all samples are changing, with the probability of good actions increasing, their advantage decreases, so the pull up strength to similar bad action will gradually vanish, raising an important question about convergence behavior: how do interdependent updates affect the long-term dynamics of training?

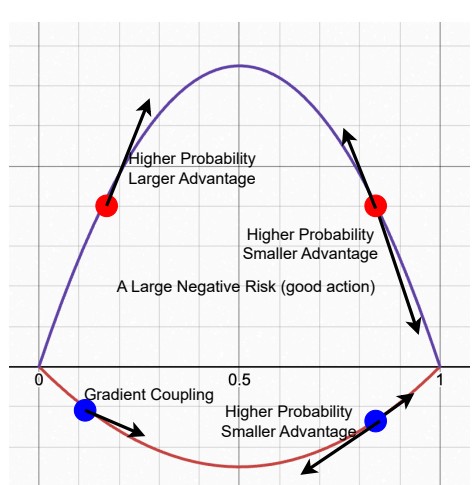

Considering the co-evolution of a good action $S_1$ (probability $p_1$, expected advantage $A_1 > 0$) and a similar flawed action $S_2$ (probability $p_2$, expected advantage $A_2 < 0$). If $|A_1| \gg |A_2|$, the strong positive advantage of $S_1$ can leak and create a positive push on $S_2$, causing both probabilities to rise initially. Suppose the gradient update on sample $S_1$ induces an additional advantage $\delta A_1$ on sample $S_2$, and vice versa, with $\delta A_2$ influencing $S_1$. Then, the effective advantage of $S_1$ becomes $A_1 + \delta A_2$, while that of $S_2$ becomes $A_2 + \delta A_1$.

Figure 4: The change of advantage

Then it is clear that

$$A_1 + \delta A_2 \geq A_2 + \delta A_1$$

So it is natural that the probability gap ($p_1 - p_2$) will be widen. At first, this is mainly because the probability of $p_1$ increases faster than $p_2$, the flawed action's probability $p_2$ will only begin to decrease after the good action's probability $p_1$ becomes sufficiently high to generate a suppressive effect that overcomes the gradient coupling. We show in Appendix A.3 that it might require $p_1$ to be very large because there could be several positive sample pushing the negative sample.

This means that some bad actions might be enhanced during training until the probability of good actions converge to a high level, as we show in Figure 5, when the training goes, the probability of some bad actions decrease, but the consistency of some other bad actions might increase, and the consistency of bad actions will start to decrease when the consistency of positive actions are relatively high. But there still exists some bad actions with high consistency, so some flawed actions might be reinforced during the learning process which brings suboptimal performance.

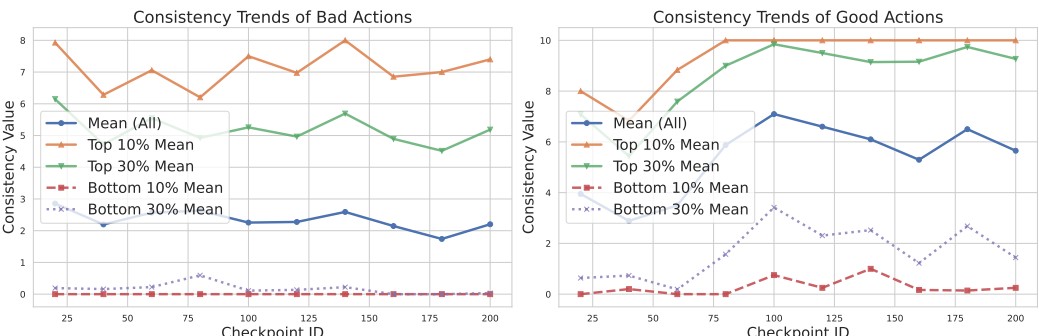

Figure 5: The dynamic of consistency in GRPO, we can observe that the consistency of some high probability bad actions increases with training. Action labels are determined by the consistency of 5 judgments from Deepseek V3.

One most common mistake with small risk should be repeated actions, we conduct experiments by using the checkpoints to generate some trajectories and we choose those repeated actions and observe the confidence of the agent. We can observe from Figure 1 that, as the training goes, the model's confidence on repeat action does not decrease. Also the percentage of those high consistency repetitions (5 times of repetition in 10 trials) is even increasing.

## 4 METHODOLOGY

Our analysis demonstrates that gradient coupling is the primary obstacle to effective agent training. To overcome this, our goal is to weaken the influence $\delta$ between samples, ensuring that good actions are reinforced while flawed ones are suppressed, i.e., $A_1 + \delta A_2 > 0$ and $A_2 + \delta A_1 < 0$, where $A_1$ and $A_2$ denote the advantages of the good and bad actions, respectively. Those methods with step level reward directly changes $A_2$ by adding extra reward, bu directly enlarging $|A_2|$, the influence caused by other samples will be relatively weaken, so it could less influenced by $A_1$. But this does not directly solve this problem because $A_1$ might also be enhanced, can we try to reduce the gradient coupling and train the model with a smaller $\delta$.

Prior work, such as Deng et al. (2025a), attempts to mitigate this interference by identifying specific tokens in negative samples and applying penalization to limit their influence on positive samples. However, this approach is not well-suited to agent-based reinforcement learning. In mathematical reasoning tasks, harmful similarity often stems from shared logical transition tokens, which can be selectively penalized without disrupting core reasoning. In contrast, in agent tasks, similarity arises primarily from overlapping action sequences and shared structures in the detailed reasoning (e.g., planning or environmental interaction steps) as illustrated in Figure 9. Therefore, weaken the penalization of these components could harm performance by acting like a removal of the negative gradient as we show in Table 3

### 4.1 WEAKEN THE GRADIENT COUPLING

As shown in Deng et al. (2025a), the gradient coupling in GRPO can be expressed as:

$$\sum_{k=1}^{|\boldsymbol{y}_i^+|} \sum_{k'=1}^{|\boldsymbol{y}_j^-|} \alpha_{k,k'} \cdot \langle \boldsymbol{h}_{\boldsymbol{x},y_{i,<k}^+}, \boldsymbol{h}_{\boldsymbol{x},y_{j,<k'}^-} \rangle, \tag{2}$$

where $\alpha_{k,k'}$ denotes a token-level similarity weight based on prediction error, and $\boldsymbol{h}_{\boldsymbol{x},y_{<k}}$ represents the hidden state embedding at position $k$ conditioned on input $\boldsymbol{x}$ and preceding tokens. This formulation reveals that gradient interference between samples arises primarily from similarities in their internal representations—specifically, when the hidden embeddings of different samples are aligned in vector space. In practice, for two distinct samples $(x_1, y_1)$ and $(x_2, y_2)$, if their inputs or reason-

ing patterns are semantically or structurally similar, their hidden states become correlated, leading to strong gradient coupling. As a result, updates intended for one sample inadvertently affect the policy for another. Therefore, to mitigate this undesirable interaction, we argue that it is essential to disentangle the embeddings of different samples during training.

For samples sharing similar label (both good actions or bad actions), the gradient coupling can actually helps to converge faster. However, when two similar samples have divergent outcomes, one leading to success and the other to failure—it becomes critical to distinguish their representations to avoid harmful gradient interference. In such cases, conflating their embeddings can lead to contradictory updates, flawed ones might be inadvertently reinforced.

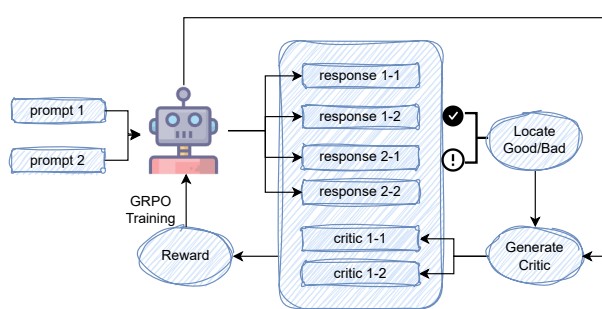

Figure 6: Training the model as a critic

This motivates our core proposal:generative classification disentanglement (**GCD**), specifically we train the actor model to simultaneously act as a classifier. As illustrated in Figure 6, we introduce an auxiliary task where the model learns to classify whether a given action is good or bad. By supervising the model on outcome labels, we force it to learn discriminative representations. Specifically, for two input-output pairs $(x_1, y_1)$ and $(x_2, y_2)$, where $y_1$ is successful and $y_2$ is not, the classification objective pulls their hidden embeddings apart in representation space. This separation ensures distinct predictions and reduces spurious gradient coupling between samples of differing quality. By explicitly decoupling good and bad behaviors in the embedding space, our approach mitigates cross-sample interference and promotes more stable and accurate policy learning in long-horizon agent tasks.

Specifically, our overall training objective is:

$$\mathcal{L} = \mathcal{L}_{\text{GRPO}} + \mathcal{L}_{\text{GCD}} \tag{3}$$

where $\mathcal{L}_{\text{GCD}}$ is a GRPO-style loss applied to a classification task. While $\mathcal{L}_{\text{GRPO}}$ optimizes the agent's actions based on task success, $\mathcal{L}_{\text{GCD}}$ optimizes the agent's classification correctness. However, as show in Garcin et al. (2025) sharing the backbone of actor and critic model in PPO could hurt the performance due to task conflict. To decline the gradient conflicts between two task, we use generative classification, and we show that it conflicts with the agent task less in Appendix B

Inspired by Feng et al. (2025), if the current observation occurs in following steps, this means that this step might take a wrong step, because it means this step may be useless. Otherwise if the current step is the last one, it means that it directly lead to success, so we recognize it as correct. Then we prompt the actor model to judge is the action a good one and allocate reward to conduct GRPO training. However, in this way we can hardly get enough positive samples, so when conduct training, the agent can easily get a high score by simply predicting all responses as negative. Therefore, we also use DeepSeek V3 to judge is it a good action then use the correct ones as training sample.

## 4.2 ESCAPING THE "DANGER ZONE" WITH PROMPT-BASED CORRECTION

Our critic-based training weakens gradient coupling, but it is worth noting that we can not entirely eliminate it. As established, this residual coupling is most dangerous when a flawed action's probability is high (the "Danger Zone"). In this regime, the action's weak self-correction mechanism is easily overpowered. To address this, we introduce a complementary strategy: prompt-based correction. During training, we collect the critiques generated by the model itself, which highlight the specific mistakes it is prone to making. We then synthesize these common errors and inject them as explicit instructions into the prompt for subsequent tasks.

This serves as a powerful, targeted intervention to drag the probability of specific flawed actions out of the "Danger Zone' and into the "Safe Regime." Once the probability is low, the natural self-

correction mechanism, can effectively take over and continue to suppress the flawed behavior during reinforcement learning.

## 5 EXPERIMENTS

### 5.1 MAIN RESULTS

Table 1: Performance of our proposed method. We show the performance when combined with GRPO, GiGPO and RLVMR. Following Zhang et al. (2025b), we split the data into three levels, seen task variants and categories (L0), unseen task with seen categories (L1) and unseen task with unseen categories (L2)

| Model | Method | ALFWorld | | | | ScienceWorld | | | |
|---|---|---|---|---|---|---|---|---|---|
| | | L0 | L1 | L2 | mean | L0 | L1 | L2 | mean |
| Qwen2.5-1.5B | Vanilla | 11.3 | 13.7 | 10.2 | 11.7 | 1.2 | 0.8 | 0.8 | 0.9 |
| | PPO | 89.0 | 85.9 | 74.2 | 83.0 | 62.7 | 52.3 | 38.2 | 51.1 |
| | GRPO | 86.7 | 85.9 | 69.7 | 80.8 | 59.4 | 50.0 | 37.5 | 49.0 |
| | with GCD | 89.8 | 88.2 | 81.5 | 86.5 | 62.7 | 54.6 | 46.1 | 54.5 |
| | GiGPO | 92.9 | 88.2 | 79.8 | 87.0 | 63.2 | 55.7 | 41.4 | 53.4 |
| | with GCD | 94.8 | 91.5 | 83.9 | 90.1 | 67.9 | 57.8 | 47.6 | 57.8 |
| | RLVMR | 85.1 | 86.7 | 72.6 | 81.5 | 63.5 | 53.7 | 40.4 | 52.5 |
| | with GCD | 87.8 | 88.2 | 78.1 | 84.7 | 65.7 | 59.3 | 42.9 | 56.0 |
| Qwen2.5-7B | Vanilla | 23.1 | 28.5 | 27.0 | 26.2 | 7.8 | 11.3 | 6.3 | 8.5 |
| | PPO | 91.6 | 92.9 | 85.8 | 90.1 | 67.9 | 60.1 | 42.7 | 56.9 |
| | GRPO | 92.3 | 91.7 | 85.1 | 89.7 | 72.4 | 59.3 | 41.4 | 57.7 |
| | with GCD | 93.4 | 92.8 | 89.8 | 92.0 | 77.3 | 67.0 | 46.1 | 63.5 |
| | GiGPO | 92.8 | 92.9 | 91.4 | 92.4 | 75.0 | 64.8 | 47.6 | 62.5 |
| | with GCD | 94.4 | 96.0 | 92.4 | 94.3 | 74.2 | 67.9 | 51.3 | 64.5 |
| | RLVMR | 92.3 | 93.6 | 86.7 | 90.9 | 76.3 | 65.6 | 42.9 | 61.6 |
| | with GCD | 94.4 | 95.7 | 92.3 | 94.1 | 76.3 | 66.3 | 48.4 | 63.7 |

We mainly conduct experiments on ALFWorld (Shridhar et al., 2020) and ScienceWorld (Wang et al., 2022), which focus on text-based scientific experimentation. We conduct cold start using the same training data of (Zhang et al., 2025b), and we train the model for 200 epochs.

We conduct experiments with Qwen2.5-1.5B and Qwen2.5-7B, and we first conduct cold start and train the model with RL algorithms like GRPO (Shao et al., 2024), GiGPO (Feng et al., 2025) and RLVMR (Zhang et al., 2025b), and we add generative classification disentanglement (**GCD**) to those methods to show how much can disentangling the embedding of similar samples help the performance. We also show the performance when we directly conduct RL without cold start in Appendix B, and we discuss the time consumption in Appendix B.1.

We present the result in Table 1, the result shows that when we use cold start, the performance on in domain test data is actually very close between different methods. Vanilla GRPO and GiGPO performs similarly, methods with step level rewards like GiGPO, RLVMR and our method also only brings marginal help especially on Qwen 2.5-7B, this is mainly because with cold start, most bad behaviours has been eliminated, so during RL training the bad behaviours will be further weakened, so those methods does not help that much.

However, when test on out domain data our method greatly helps, this is mainly because for those out domain data, the bad actions can not be directly punished, it can only be indirectly influenced by other similar sample gradients, and our method cluster the embedding of bad actions which can help to punish bad actions while reduce the influence of other similar good actions. Also, our experimental result seems to be higher than the one show in Zhang et al. (2025b), this is mainly because they simply train the model fro 100 epochs, and we observe that 100 epochs is not enough for convergence, so we train it for 200 epochs.

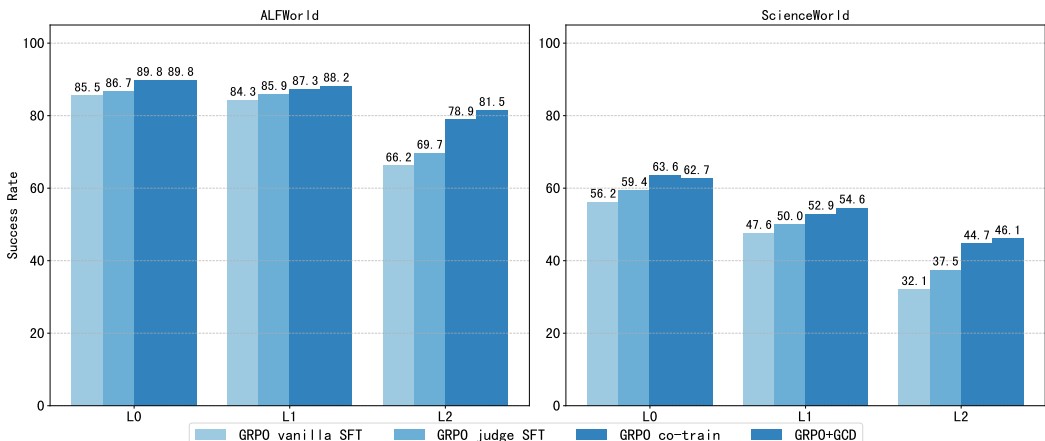

Figure 7: The performance with different cold start and setting. GRPO vanilla SFT means we conduct cold start with only the agent task, GRPO judge SFT means we conduct cold start with generative classification task. GRPO co-train means when conduct GRPO, we use the auxiliary task, and GRPO GCD means we also summarize the critics and insert it into the prompt

## 5.2 ABLATION STUDY

We claim that by training the model as a generative judge to classify is the action a good one to separate the embedding of good and bad actions which could effectively help reduce the gradient coupling and improve the performance. Therefore, when conducting cold start, if we add generative judge data, it can also effectively improve the performance as we show in Figure 7 that cold start with generative judge data helps the performance. Also in Figure 7 we show that training the actor as a classifier greatly helps the performance, and by summarizing the generated critic into a coherent suggestion for the actor also helps.

## 5.3 THE GRADIENT COUPLING BETWEEN SAMPLES

As we claimed before, our method can effectively separate the embedding between positive and negative samples, following Equation 2, we calculate the influence on of one token on other samples and average the influence of tokens to show the influence between samples. We use the gap of influence between same class samples and the influence between different class to show how much it take advantage of the gradient coupling and eliminate its negative effect, the result is shown in Figure 8. It shows that our method effectively separate the embedding of positive and negative samples, the gap is much larger than vanilla GRPO and GiGPO.

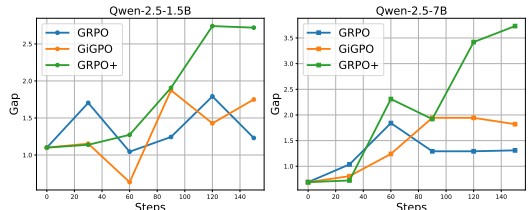

Figure 8: The change of gradient coupling during the training process

## 6 CONCLUSION

In this paper, we reveal that during GRPO, the expected advantage of flawed actions should be negative even considering the squeezing effect. Then we show that the reason why flawed action will be mistakenly enhanced mainly due to gradient coupling, which means that the gradient of positive actions might mistakenly increase the probability of bad action due to the similarity of the actions. Then, we propose to co-train the actor as a generative judge to disentangle the embedding of good and bad actions.

REPRODUCIBILITY STATEMENT

We provide our code in `https://anonymous.4open.science/r/RL_GCD-E562`, we show the experimental setting in Appendix B, and the prompts are also shown in Appendix A.4.

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

## LLM USAGE

This paper simply uses LLM to polish paragarphs, adjust grammars and help to generate code for drawing pictures in the paper.

## A PROOFS

### A.1 PROOF OF LEMMA 3.1

Given a policy $\pi_\theta$, with probability $p$ it would fail completing the task and get reward $0$, otherwise success with reward $1$. However, with probability $q$, it will take a mistake, and the mistake could enlarge the risk of failing by $r$.

In this way, the success and fail probability could be represented as follows

$$\begin{cases} (1-q)\cdot(1-p), & \text{no mistake and success} \\ (1-q)\cdot p, & \text{no mistake but fail} \\ q\cdot(1-p-r), & \text{mistake but success} \\ q\cdot(p+r), & \text{mistake and fail} \end{cases} \tag{4}$$

Then, we can calculate that,

$$\begin{aligned} \mathbb{P}(success) &= (1-q)\cdot(1-p) + q\cdot(1-p-r) = 1-p-qr, \\ \mathbb{P}(fail) &= (1-q)\cdot p + q\cdot(p+r) = p+qr. \end{aligned} \tag{5}$$

In GRPO, the advantage is calculated as $A_i = \frac{s_i - \mathbb{E}(s)}{\text{std}(s)}$, where $A_i$ means the advantage, $s_i$ is the calculated reward score. Based on Equation 5, we have that $\mathbb{E}(a) = 1\cdot\mathbb{P}(success) + 0\cdot\mathbb{P}(fail) = 1-p-qr$

Then, we have that

$$A_i = \begin{cases} \frac{p+qr}{\text{std}(r)}, & a_i = 1 \\ \frac{p+qr-1}{\text{std}(r)}, & a_i = 0 \end{cases} \tag{6}$$

Then the expected advantage about the mistake is (we ignore the std),

$$\begin{aligned} A_{mistake} &= q\cdot(1-p-r)\cdot(p+qr) + q\cdot(p+r)\cdot(p+qr-1) \\ &= (q-pq-qr)\cdot(p+qr) + (pq+qr)\cdot(p+qr-1) \\ &= (q-pq-qr+pq+qr)\cdot(p+qr) - pq-qr \\ &= pq + q^2r - pq - qr = qr(q-1) \end{aligned} \tag{7}$$

This means that unless $q=1$ or $r=0$, the advantage will be negative, which means unless the mistake will happen with probability $1$ or the mistake will not lead to any risk of failure, it will be discouraged.

In contrast, if $r < 0$, which means that the action will reduce the risk of failure, then it will be encouraged unless the probability of taking the action is already $1$.

## A.2 PROOF OF THEOREM 3.2

However, due to the squeezing effect, the negative gradient might encourage the mistake. Following Ren & Sutherland (2024), we also use logistic regression to show the squeezing effect.

Consider a simple V-class logistic regression problem where each high-dimensional input data $x$ is converted to a length-$d$ feature vector via a deep neural network $\pi$. The model uses a linear read-out layer $\boldsymbol{w} \in \mathbb{R}^{d \times V}$ to convert the feature vector to logits $z = \boldsymbol{w}^T \cdot \pi(x)$ and then generate the probability prediction vector $p$ using a Softmax head. We consider a common cross-entropy loss function for each input pair $(x, y)$. In summary, we have

$$\mathcal{L}_{CE}(\boldsymbol{p}^t, y) = -\boldsymbol{e}_y \log(\boldsymbol{p}^t); \quad \boldsymbol{p}^t = \text{Softmax}(\boldsymbol{z}^t); \quad \boldsymbol{z}^t = (\boldsymbol{w}^t)^T \pi(\boldsymbol{x}),$$

where $t$ is the index of the step during training and $\boldsymbol{e}_y$ is a length-V one-hot vector determined by the ground truth label y. To simplify our analysis, we assume a fixed $\pi$ and only update the parameters of the read-out layer $\boldsymbol{w}$ using stochastic gradient descent:

$$\boldsymbol{w}^{t+1} = \boldsymbol{w}^t - \eta \Delta_{\boldsymbol{w}} \mathcal{L} = \boldsymbol{w}^t - \eta \pi(\boldsymbol{x})(\boldsymbol{p}^t - \boldsymbol{e}_y)^T, \tag{8}$$

where $\eta$ is the learning rate. In GRPO, there are multiple positive and negative gradients, and the magnitude of positive and negative gradient is determined by the advantage $A$, so we consider

$$\boldsymbol{w}^{t+1} = \boldsymbol{w}^t - \eta \Delta_{\boldsymbol{w}} \mathcal{L} = \boldsymbol{w}^t - \sum_y \left( A_y \cdot \eta \pi(\boldsymbol{x})(\boldsymbol{p}^t - \boldsymbol{e}_y)^T \right). \tag{9}$$

Therefore, we have that,

$$\boldsymbol{z}^{t+1} = (\boldsymbol{w}^{t+1})^T \pi(\boldsymbol{x})$$

$$= \left( \boldsymbol{w}^t - \sum_y \left( A_y \cdot \eta \pi(\boldsymbol{x})(\boldsymbol{p}^t - \boldsymbol{e}_y)^T \right) \right)^T \pi(\boldsymbol{x})$$

$$= \boldsymbol{w}^T \pi(\boldsymbol{x}) - \left( \sum_y \left( A_y \cdot \eta \pi(\boldsymbol{x})(\boldsymbol{p}^t - \boldsymbol{e}_y)^T \right) \right)^T \pi(\boldsymbol{x})$$

$$= \boldsymbol{z}^t - \eta ||\pi(\boldsymbol{x})||_2^2 \cdot \left( \sum_y A_y (\boldsymbol{p}^t - \boldsymbol{e}_y) \right)$$

If we consider an action $i$ with probability $\hat{q}$ and expected advantage $\frac{\hat{A}}{std}$.

$$\boldsymbol{z}_i^{t+1} = (\boldsymbol{w}^{t+1})^T \pi(\boldsymbol{x}) = \boldsymbol{z}_i^t - \eta ||\pi(\boldsymbol{x})||_2^2 \cdot \left( \sum_y A_y (\boldsymbol{p}_i^t - \boldsymbol{e}_{y,i}) \right)$$

$$= (\boldsymbol{w}^{t+1})^T \pi(\boldsymbol{x}) = \boldsymbol{z}_i^t - \eta ||\pi(\boldsymbol{x})||_2^2 \cdot \left( \sum_{y=i} A_y (\hat{q} - 1) + \sum_{y \neq i} A_y (\hat{q}) \right)$$

$$= \boldsymbol{z}_i^t - \eta ||\pi(\boldsymbol{x})||_2^2 \left( n \frac{1}{std} \hat{A}(\hat{q} - 1) - n \frac{1}{std} \hat{A}\hat{q} \right) \tag{10}$$

$$= \boldsymbol{z}_i^t - \eta ||\pi(\boldsymbol{x})||_2^2 \cdot n \frac{1}{std} \left( \hat{A}(\hat{q} - 1) - \hat{A}\hat{q} \right)$$

$$= \boldsymbol{z}_i^t + \eta ||\pi(\boldsymbol{x})||_2^2 \cdot n \frac{1}{std} \hat{A}$$

The second equation is because $\frac{\hat{A}}{std}$ is expected advantage with one action, so when conducting $n$ actions, the advantage should be $n \frac{\hat{A}}{std}$, and the summation of all advantages should be 0. Therefore, we can have that for the action with probability $q$ and risk $r$ as $\boldsymbol{z}_*^{t+1}$, then the advantage is already calculated to be $qr(q - 1)$, then

$$\boldsymbol{z}_*^{t+1} = \boldsymbol{z}_i^t + \eta||\pi(\boldsymbol{x})||_2^2 \cdot n\frac{1}{std} \cdot qr(q-1) = \boldsymbol{z}_i^t + \eta||\pi(\boldsymbol{x})||_2^2 \cdot n\frac{1}{std}qr(q-1)$$

Now consider a simple case with 3 different actions. And let the first action be the one with probability $q$ and risk $r$, the second action with probability $\hat{q}$ and risk $\hat{r}$, then for the third action, the probability is $1 - q - \hat{q}$. From previous assumption, we require the expected probability of failure to be $p$ with the second and third action, so the risk of the third action is $\frac{\hat{q}\cdot\hat{r}}{q+\hat{q}-1}$.

For a action with probability $q$ and risk $r$, we can calculate the advantage as follows:

$$\begin{aligned} A &= \hat{q} \cdot (1 - p - \hat{r}) \cdot (p + qr) + \hat{q} \cdot (p + \hat{r}) \cdot (p + qr - 1) \\ &= (p + qr)\hat{q}(1 - p - \hat{r} + p + \hat{r}) - \hat{q}(p + \hat{r}) \\ &= \hat{q}(p + qr) - \hat{q}(p + \hat{r}) \\ &= p\hat{q} + q\hat{q}r - p\hat{q} - \hat{q}\hat{r} \\ &= \hat{q}(qr - \hat{r}) \end{aligned}$$

Therefore we have the following,

$$A_i = \begin{cases} qr(q-1), & i = 1 \\ \hat{q}(qr - \hat{r}), & i = 2 \\ (1 - q - \hat{q})(qr - \frac{\hat{q}\cdot\hat{r}}{q+\hat{q}-1}), & i = 3 \end{cases}$$

Then, based on Equation 10, we know that,

$$\boldsymbol{z}_1^{t+1} = \boldsymbol{z}_1^t + \eta||\pi(\boldsymbol{x})||_2^2 \cdot n\frac{1}{std}qr(q-1)$$

$$\boldsymbol{z}_2^{t+1} = \boldsymbol{z}_2^t + \eta||\pi(\boldsymbol{x})||_2^2 \cdot n\frac{1}{std}\hat{q}(qr - \hat{r})$$

$$\boldsymbol{z}_3^{t+1} = \boldsymbol{z}_3^t + \eta||\pi(\boldsymbol{x})||_2^2 \cdot n\frac{1}{std}(1 - q - \hat{q})(qr - \frac{\hat{q}\cdot\hat{r}}{q+\hat{q}-1})$$

Then, we can have that

$$p_1^{t+1} = \frac{\exp(\boldsymbol{z}_1^{t+1})}{\exp(\sum_j \boldsymbol{z}_j^{t+1})} = \frac{\exp(z_1^t + C \cdot A_1)}{\sum_j \exp(z_j^t + C \cdot A_j)}$$

$$p_1^t = \frac{\exp(z_1^t)}{\sum_j \exp(z_j^t)}$$

And we can calculate does the probability $p_1^{t+1}$ increase by

$$\begin{aligned} \frac{p_1^{t+1}}{p_1^t} &= \frac{\exp(z_1^t + C \cdot A_1)}{\sum_j \exp(z_j^t + C \cdot A_j)} \cdot \frac{\sum_j \exp(z_j^t)}{\exp(z_1^t)} \\ &= \frac{\exp(z_1^t)}{\sum_j \exp(z_j^t + C \cdot (A_j - A_1))} \cdot \frac{\sum_j \exp(z_j^t)}{\exp(z_1^t)} \\ &= \frac{\sum_j \exp(z_j^t)}{\sum_j \exp(z_j^t + C \cdot (A_j - A_1))} \end{aligned}$$

Therefore, to determine is $\frac{p_1^{t+1}}{p_1^t} > 1$, we need to know is

$$\exp(z_2^t + C \cdot (A_2 - A_1)) + \exp(z_3^t + C \cdot (A_3 - A_1)) > \exp(z_2^t) + \exp(z_3^t)$$

Then we can calculate as follows,

$$\begin{aligned} &\exp(z_2^t + C \cdot (A_2 - A_1)) + \exp(z_3^t + C \cdot (A_3 - A_1)) - \exp(z_2^t) + \exp(z_3^t) \\ &= (\exp(C \cdot (A_2 - A_1)) - 1) \cdot \exp(z_2^t) + (\exp(C \cdot (A_3 - A_1)) - 1) \cdot \exp(z_3^t) \end{aligned}$$ (11)

As we know that $p_2 = \hat{q}$ and $p_3 = 1 - q - \hat{q}$, and let $\delta = \sum_j (z_j^y)$, we can have that $\exp(z_2^t) = \delta \cdot \hat{q}$, and $\exp(z_3^t) = \delta \cdot (1 - q - \hat{q})$

Since we only care about is it larger or smaller than 0, we ignore some constant and have.

$$
\begin{aligned}
&\exp(z_2^t + C \cdot (A_2 - A_1)) + \exp(z_3^t + C \cdot (A_3 - A_1)) - \exp(z_2^t) + \exp(z_3^t) \\
&= (\exp(C \cdot (A_2 - A_1)) - 1) \cdot \hat{q} + (\exp(C \cdot (A_3 - A_1)) - 1) \cdot (1 - q - \hat{q}) \\
&\leq e \cdot C(A_2 - A_1)\hat{q} + e \cdot C(A_3 - A_1) \cdot (1 - q - \hat{q}) \\
&= e \cdot C \left(\hat{q}A_2 - \hat{q}A_1 + A_3 - A_1 - qA_3 + qA_1 - \hat{q}A_3 + \hat{q}A_1\right) \\
&= e \cdot C \left(\hat{q}(A_2 - A_3) + (q - 1)(A_1 - A_3)\right)
\end{aligned}
\tag{12}
$$

$$
\begin{aligned}
A_2 - A_3 &= \hat{q}(qr - \hat{r}) - (1 - q - \hat{q})(qr - \frac{\hat{q} \cdot \hat{r}}{q + \hat{q} - 1}) \\
&= q\hat{q}r - \hat{q}\hat{r} - (1 - q - \hat{q})qr - \hat{q}\hat{r} \\
&= 2q\hat{q}\hat{r} + q^2 r - qr - 2\hat{q}\hat{r}
\end{aligned}
$$

$$
\begin{aligned}
A_1 - A_3 &= qr(q - 1)(qr - \frac{\hat{q} \cdot \hat{r}}{q + \hat{q} - 1}) \\
&= 2q^2 r - 2qr + q\hat{q}r - \hat{q}\hat{r}
\end{aligned}
$$

Then, we can calculate

$$
\begin{aligned}
&\hat{q}(A_2 - A_3) + (q - 1)(A_1 - A_3) \\
&= 2q\hat{q}^2\hat{r} + \hat{q}q^2 r - \hat{q}qr - 2\hat{q}^2\hat{r} + 2q^3 r - 2q^2 r + q^2\hat{q}r - q\hat{q}\hat{r} - 2q^2 r + 2qr - q\hat{q}r + \hat{q}\hat{r} \\
&= \hat{q}^2\hat{r}(2q - 2) + \hat{q}(q^2 r - qr) + \hat{q}(q^2 r - qr) - q\hat{q}\hat{r} + \hat{q}\hat{r} + \delta \quad (\delta = 2q^3 r - 4q^2 r + 2qr) \\
&= 2\hat{q}^2\hat{r}(q - 1) + \hat{q}(qr)(q - 1) + \hat{q}qr(q - 1) - \hat{q}\hat{r}(q - 1) + \delta \\
&= (2\hat{q}^2\hat{r} + 2\hat{q}qr - \hat{q}\hat{r})(q - 1) + \delta
\end{aligned}
\tag{13}
$$

Therefore, to make it smaller than 0, we require

$$
\begin{aligned}
(2\hat{q}^2\hat{r} + 2\hat{q}qr - \hat{q}\hat{r})(q - 1) + \delta &<= 0 \\
2\hat{q}^2\hat{r} + 2\hat{q}qr - \hat{q}\hat{r} &\geq -\frac{2q^3 r - 4q^2 r + 2qr}{q - 1} = 2qr(1 - q)
\end{aligned}
\tag{14}
$$

In this way, we show that in some cases, even if the action can have negative effect on the performance, its probability would be encouraged instead of discouraged (an increasing $q$). Also as we calculated in Equation 7, the advantage is $qr(q - 1)$, if we consider $r$ as a constant, then the left side of 14 will increase, while the right side might even decrease considering that $q(1 - q)$ will decrease with $q$ if $q$ is larger than $\frac{1}{2}$.

Also, we can show that $z_3^{t+1} - z_1^{t+1}$

$$
\begin{aligned}
z_3^{t+1} - z_1^{t+1} &= z_3^t - z_1^t + \eta \|\pi(\boldsymbol{x})\|_2^2 \cdot n \frac{1}{std} \left((1 - q - \hat{q})(qr - \frac{\hat{q} \cdot \hat{r}}{q + \hat{q} - 1}) - qr(q - 1)\right) \\
\Delta z^{t+1} - \Delta z^t &= \eta \|\pi(\boldsymbol{x})\|_2^2 \cdot n \frac{1}{std} \left((1 - q - \hat{q})(qr - \frac{\hat{q} \cdot \hat{r}}{q + \hat{q} - 1}) - qr(q - 1)\right)
\end{aligned}
\tag{15}
$$

Therefore, we have that

$$
\begin{aligned}
\text{sign}(\Delta z^{t+1} - \Delta z^t) &= \text{sign} \left((1 - q - \hat{q})(qr - \frac{\hat{q} \cdot \hat{r}}{q + \hat{q} - 1}) - qr(q - 1)\right) \\
&= \text{sign} \left((1 - q - \hat{q})qr + \hat{q} \cdot \hat{r} + qr(q - 1)\right) \\
&= \text{sign} \left((2 - 2q - \hat{q})qr + \hat{q}\hat{r}\right)
\end{aligned}
\tag{16}
$$

It is clear that $2 - 2q - \hat{q} = (1 - q - \hat{q}) + (1 - q) \geq 0$

## A.3 PROOF OF THEOREM A.1

Consider two different sample $(x_1, y_1)$, $(x_2, y_2)$, sample 1 holds probability $q_1$ and risk $r_1$ while sample 2 holds probability $q_2$ and risk $r_2$. Also we assume that the logits before softmax is $z_1$ and $z_2$, we assume that

$$\sum_{y!=y_1} \exp(z_1[y]) = \sum_{y!=y_2} \exp(z_2[y]) = C \tag{17}$$

Therefore, when conducting softmax, we have that

$$q_1 = \frac{\exp(z_1[y_1])}{\exp(z_1[y_1]) + C}, \quad q_2 = \frac{\exp(z_2[y_2])}{\exp(z_2[y_2]) + C} \tag{18}$$

For sample 1, we have Advantage $A_1 = q_1 r_1 (q_1 - 1)$, and for sample 2 $A_2 = q_2 r_2 (q_2 - 1)$, then with the influence $\delta$, then the advantage for sample 2 could be represented as $\hat{A}_2 = A_2 + \delta A_1 = q_2 r_2 (q_2 - 1) + \delta q_1 r_1 (q_1 - 1)$. We letting $r_1 = -\xi r_2$, and in the following we use $r$ instead of $r_2$ for simplicity, and we want $\hat{A}_2 <= 0$, note that $r > 0$, then

$$\hat{A}_2 = A_2 + \delta A_1 = q_2 r (q_2 - 1) - \xi \delta q_1 r (q_1 - 1) \leq 0$$
$$q_2 (q_2 - 1) \leq \xi \delta q_1 (q_1 - 1)$$
$$\frac{q_2 (q_2 - 1)}{q_1 (q_1 - 1)} \geq \xi \delta \tag{19}$$
$$\frac{q_2 (1 - q_2)}{q_1 (1 - q_1)} \geq \xi \delta$$

If we assume that the logit of other $z$ stays unchanged, considering that,

$$q_1 = \frac{\exp(z_1[y_1])}{\exp(z_1[y_1]) + C}, \quad q_2 = \frac{\exp(z_2[y_2])}{\exp(z_2[y_2]) + C}$$
$$1 - q_1 = \frac{C}{\exp(z_1[y_1]) + C}, \quad 1 - q_2 = \frac{C}{\exp(z_2[y_2]) + C} \tag{20}$$

Then we have that

$$\frac{q_2(1 - q_2)}{q_1(1 - q_1)} = \frac{\exp(z_2[y_2])}{\exp(z_2[y_2]) + C} \cdot \frac{C}{\exp(z_2[y_2]) + C} \cdot \frac{\exp(z_1[y_1]) + C}{\exp(z_1[y_1])} \cdot \frac{\exp(z_1[y_1]) + C}{C}$$
$$= \frac{h_2}{h_2 + C} \cdot \frac{C}{h_2 + C} \cdot \frac{h_1 + C}{h_1} \cdot \frac{h_1 + C}{C} \tag{21}$$
$$= \frac{h_2 \cdot (h_1 + C)^2}{h_1 \cdot (h_2 + C)^2} \geq \xi \delta$$

If we assume that $h_2 \leq h_1$, which means $q_2 \leq q_1$, then we have that

$$\xi \delta \leq \frac{h_2 \cdot (h_1 + C)^2}{h_1 \cdot (h_2 + C)^2}$$
$$\leq \left( \frac{h_1 + C}{h_2 + C} \right)^2 \leq \left( \frac{h_1}{h_2} \right)^2 \tag{22}$$
$$= \exp(z_1 - z_2)^2$$

So, we require $z_1 - z_2 \geq \frac{1}{2} \ln \xi \delta$.

If we assume that the relationship between the advantage and the logit is linear, which means that with advantage $A$, then $\delta z = \eta \cdot A$,

Then we have that $z_1 - z_2 = \hat{z_1} - \hat{z_2} + \eta \sum_i \left( A_1^i + \delta A_2^i - A_2^i - \delta A_1^i \right)$, where $\hat{z_1}$ represents the original value of the logit before GRPO, and $\sum_i$ represents all the advantage of the GRPO steps.

Also we have that for all $i$, $A_2^i + \delta A_1^i > 0$, which means $A_2^i > -\delta A_1^i$.

Therefore,

$$
\begin{aligned}
\frac{1}{2} \ln \xi \delta \leq z_1 - z_2 &= \hat{z_1} - \hat{z_2} + \eta \sum_i \left( A_1^i + \delta A_2^i - A_2^i - \delta A_1^i \right) \\
&= \hat{z_1} - \hat{z_2} + \eta \sum_i \left( (1 - \delta) A_1^i + (\delta - 1) A_2^i \right) \\
&\leq \hat{z_1} - \hat{z_2} + \eta \sum_i \left( (1 - \delta) A_1^i + (\delta - 1)(-\delta A_1^i) \right) \\
&\leq \hat{z_1} - \hat{z_2} + \eta \sum_i \left( (1 - \delta^2) A_1^i \right) \\
&= \leq \hat{z_1} - \hat{z_2} + + \Delta z_1.
\end{aligned}
\tag{23}
$$

So, we have that $\Delta z_1 \geq \frac{1}{2} \ln \xi \delta + \Delta \hat{z}$, $h_1 = \exp(\frac{1}{2} \ln \xi \delta + \Delta \hat{z}) \cdot \hat{h_1}$.

**Theorem A.1.** *For two different kind samples $(x_1, y_1)$ and $(x_2, y_2)$ with risk $r_1$ $r_2$, if we assume $r_1 = -\xi r_2$ ($r_2 > 0$), if the summation logit other actions of $x_1$ and $x_2$ are the same, and the probability of both $y_1$ and $y_2$ is greater than $0.5$. Then we have that, we require $h_1$, the logit of $y_1$ to be $\exp(\frac{1}{2} \xi \delta + \Delta \hat{z})$ times than its original value until the probability of $y_2$ start to decrease.*

## A.4 PROMPTS

---

**Prompt for generate suggestion**

**Role:** You are an Expert AI Strategist, tasked with synthesizing raw feedback to generate high-level, actionable principles for improving an agent's performance. The agent is operating in a complex environment like ALFWorld or ScienceWorld.
**Context:** You will be provided with a pre-defined theme and a list of specific advice sentences that have already been clustered under this theme.
**Your Sole Task:** Distill the entire collection of related advice into one single, overarching "Golden Rule."
This rule must be:

- High-Level: Abstract away from specific examples.
- Actionable: Provide clear guidance on what the agent should do.
- Generalizable: Be applicable to future, unseen situations related to this theme.

For each theme provided in the input data, you must perform the following steps: 1. Think Step-by-Step: First **you must think step by step and dig deep into the advices to formulate the high-level principle.** Analyze the specific advice, identify the common pattern or root cause, and build a line of reasoning toward a general rule. 2. Formulate the Rule: Based on your thinking, synthesize the advice into **one single, actionable, and generalizable Golden Rule."
ADVICE LIST: {advice list}

**Prompt for critic generation**

**Role**: You are an expert evaluator, a "Critic" tasked with judging the quality of an action taken by another agent in ALFWorld/SciWorld, a household/science environment where the agent is required with some tasks like 'put a cellphone in bed' or 'determine if metal fork is electrically conductive'

Given the question and the agent response you should judging the quality of the response

**Context:**

- **Problem:** {problem}
- **Agent Response:** {response}

The problem has not been solved in current state, if the agent considered the problem has been done before taking any action, it should be a serious error.

**Your Task:**

1. **Analyze:** In your '⟨think⟩' block, perform a step-by-step analysis.

- Consider the 'Current State' and the overall 'Problem'.
- Evaluate the thinking process of the agent and check is the thinking logical and if the 'Chosen Action' makes logical progress.
- Evaluate will the 'Chosen Action' cause serious trouble or it is obvious not a good option or it may end in a loop of actions.
- Evaluate other possible actions and analyze is there some action obviously better than the chosen one.
- You should notice that the agent is good at this task, the success rate is about 80 percent, but it will also make some small mistakes or redundant actions, so you should not only consider the bad side of the action, also consider the good side.

2. **Judge:** In the '⟨answer⟩' block, provide a score. Use '1' for a good action, and '0' if the action is obviously a poor action and there exists explicitly better actions.

3. **Propose (if applicable):** In the ⟨action⟩ block, if (and only if) the score is '0', provide the single best alternative action based on the available actions. Do not add any explanation here, only the action itself.

4. **Trajectory Analyze:** Based on previous actions and its correspondding feedback, analyze its previous actions and check is there some actions could be further improved

5. **Advise:** In the '⟨advise⟩' block, provide a single, concise sentence of feedback.

- Based on your analysis of current action and previous actions, provide advice to help the agent avoid its mistakes and enhance its advantage in the future.
- Provide a high-level strategic principle that applies to various situations, rather than a correction for this specific instance.

**Output Format:** You must follow this exact format. The '⟨action⟩' block is conditional and should only appear when the score is 0.

⟨think⟩ Your step-by-step reasoning here. This is where you will explain why the chosen action is good or bad. If it's bad, you will also explain why your proposed alternative action is better. ⟨/think⟩

⟨answer⟩
boxed{0/1}⟨/answer⟩ ⟨action⟩The single best alternative action for current observation here (only if score is 0)⟨/action⟩ ⟨advise⟩Your concise, high-level strategic advice here ⟨/advise⟩

---

**Prompt for ALFWorld**

You are an expert agent operating in the ALFRED Embodied Environment. Your task is to: {task description} You should strictly follow the guidelines below, do what the guideline suggests in your thinking steps to make better actions: {suggestions} Prior to this step, you have already taken {step count} step(s). Below are the most recent {history length} observaitons and the corresponding actions you took: {action history} You are now at step {current step} and your current observation is: {current observation} Your admissible actions of the current situation are: [{admissible actions}].

Now it's your turn to take an action. You should first reason step-by-step based on the guideline about the current situation. This reasoning process MUST be enclosed within $\langle think \rangle$ $\langle /think \rangle$ tags. Once you've finished your reasoning, you should choose an admissible action for current step and present it within $\langle action \rangle$ $\langle /action \rangle$ tags.

---

**Prompt for ScienceWorld**

You are an expert agent operating in the ScienceWorld environment, which is a text-based virtual environment centered around accomplishing tasks from the elementary science curriculum. You should strictly follow the guidelines below, do what the guideline suggests in your thinking steps to make better actions: {suggestions} Your current task is: {task description}

Prior to this step, you have already taken {step count} step(s). Below are the most recent {history length} observations and the corresponding actions you took: {action history} You are now at step {current step} and your current observation is: {current observation} Here are the actions you may take:

Current available actions: {available actions}

Now it's your turn to take an action. You should first reason step-by-step about the current situation. This reasoning process MUST be enclosed within $\langle think \rangle$ $\langle /think \rangle$ tags. Once you've finished your reasoning, you should choose an appropriate action for the current step and present it within $\langle action \rangle$ $\langle /action \rangle$ tags.

---

## B  MORE EXPERIMENTS

We use the dataset and code of RLVMR (Zhang et al., 2025b) to generate data for cold start, we use 300 trajectories as recommended in the code of RLVMR. Then we use DeepSeek-V3 to generate 100 critic data and incorporate it into the training data. When conducting cold start, we use 8 PPU with per device batch size 2 and learning rate 1e-5, we train it for 5 epochs. For reinforcement learning, we also use 8 PPUs with train data size 16 and group size 8 and learning rate 1e-6, the max steps of interaction with environment is set to 30. When conducting RL, we train Qwen2.5 1.5B for 200 epochs and 7B for 100 epochs as it converges faster

Table 2: The performance without cold start with L2

| | Method | ALFWorld | ScienceWorld |
|---|---|---|---|
| | GRPO | 48.4 | 26.6 |
| | with GCD | 52.3 | 31.2 |
| | GiGPO | 67.1 | 25.8 |
| Qwen-7B | with GCD | 69.8 | 32.4 |
| | RLVMR | 65.3 | 27.3 |
| | with GCD | 68.7 | 29.5 |

We show the performance when we use value based classifier and use the classification task and agent task to conduct cold start and the following RL in Table 3, we can observe that by conducting cold start with sone generative judge data, the performance increases. Also we show that directly conduct penalization on the advantage tokens cause similarity between different sample as show in

Table 3: The performance with different cold start. Vanilla SFT mean vanilla cold start, generative SFT means cold start with generative critic task, and value SFT means cold start with value based classifier. GRPO means GRPO with the instruct model, GRPO SFT means GRPO generative SFT as cold start, and GRPO_nthr means we add penalization to those similar tokens which would cause gradient coupling, the model is tested on ALFWorld L1

|  | vanilla SFT | generative SFT | value SFT | GRPO | GRPO_nthr |
|---|---|---|---|---|---|
| Qwen 1.5B | 61.7 | 63.4 | 59.5 | 85.9 | 82.7 |
| Qwen 7B | 65.2 | 66.7 | 63.7 | 91.7 | 89.4 |

Deng et al. (2025a) may not help in agent task. As we show in Figure 9, the tokens cause similarity in agent task is key tokens about the reasoning.

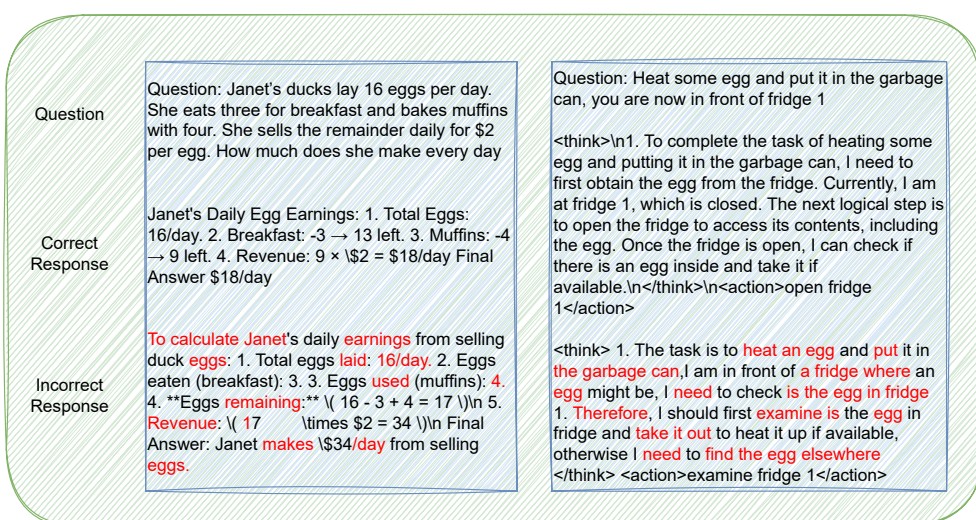

Figure 9: The tokens that cause the similarity in gradient

To train the classifier, one direct way is to add a value head to conduct prediction, but we show in Figure 10b that the two tasks are too different, their requires different embeddings and conflicting with each other leading to suboptimal performance as show in Appendix B. So we use generative judge, specifically, as show in Figure 6, when training the agent using GRPO, we can select some of the responses and let the model judge is it a good action for several times, then we use rule based reward to allocate reward to those responses and use GRPO to train it. If the model can effectively judge is it a good action, then the embedding between positive and negative samples inherent the model could be more separated and the influence between different could be weaken (small $\delta$).

## B.1    TIME CONSUMPTION

Table 4: The time consumption (hours) of different methods. Our method costs about 30% more time but it converges faster, we show the performance of vanilla at epoch 200 and GCD at epoch 150 and it performs similarly

|  | Time | | | Performance | | |
|---|---|---|---|---|---|---|
|  | GRPO | GiGPO | RLVMR | GRPO | GiGPO | RLVMR |
| vanilla | 14.25 | 9.67 | 11.16 | 69.7 | 79.8 | 72.6 |
| GCD | 18.45 | 13.32 | 16.46 | 72.5 | 78.6 | 74.1 |

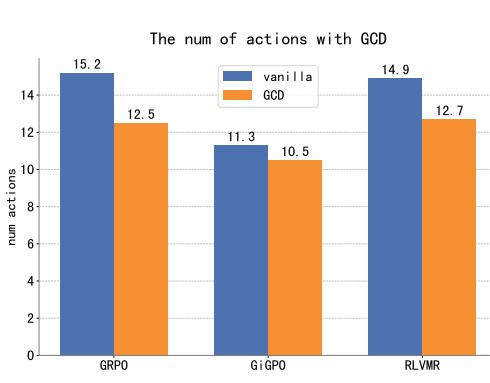

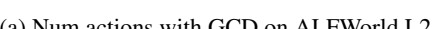

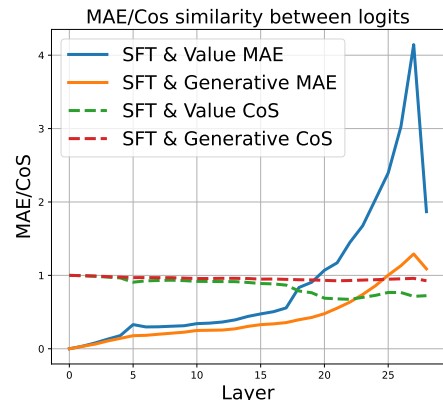

(a) Num actions with GCD on ALFWorld L2

(b) The MAE and cosine similarity of logits in each layer. SFT & Value means the difference/similarity between logits of vanilla SFT model and the value based classification model. SFT & Generative means the difference.similarity between logits of vanilla SFT model and the generative critic model.

Figure 10: The number of actions and the influence of valued head based classification

We can observe from Table 4 that our method adds about $\frac{1}{3}$ time for training, also we show the performance of vanilla at epoch 200 and GCD at epoch 150, this shows that that our method already hold similar performance with vanilla method in epoch 150, so our method indeed requires more training time, but it cost similar time to converge to similar performance and it can performs better. And Figure 11 shows that 200 epoch already converges, continually train the model does not improve the performance.

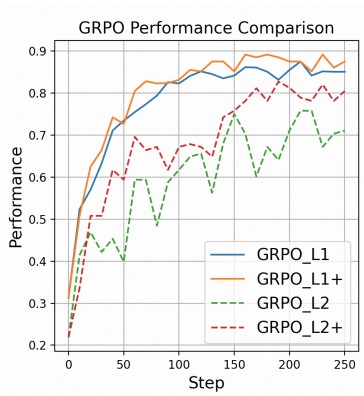

Figure 11: The learning dynamic of GRPO

