# OpenReview forum: "Rethinking Reward Miscalibration of GRPO in Agentic RL"
_ICLR.cc/2026/Conference — ICLR 2026 Conference Withdrawn Submission_

### Official Review · Reviewer_eAwJ · 2025-10-20

**Soundness:** 3
**Presentation:** 3
**Contribution:** 2
**Rating:** 4
**Confidence:** 4

**Summary:**

The paper challenges the idea that GRPO struggles due to reward miscalibration, arguing instead that the real issue is gradient coupling: shared neural representations cause updates from successful trajectories to inadvertently reinforce similar but flawed actions, especially early in training. To mitigate this, the authors propose Generative Classification Disentanglement (GCD), which adds an auxiliary task where the policy classifies actions as good or bad to disentangle their embeddings, along with lightweight prompt corrections. Experiments on ALFWorld and ScienceWorld show that GCD consistently improves performance, particularly in out-of-domain settings, highlighting that addressing representational interference is more effective than simply adjusting reward structure.

**Strengths:**

- Reframes “reward miscalibration” and identifies gradient coupling as the true failure mode, with a clear, intuitive narrative grounded in the RL context.

- I really like how the authors present their approach: the coupling diagnosis is clean, falsifiable, and measurable.

**Weaknesses:**

- the “gradient coupling” story here is a specific LLM-flavoured instance of a very common RL pain point: function-approximation spillover. Neural nets are smooth; many control problems are effectively discrete. When we push up the log-prob for a “good” behaviour, nearby representations get dragged along. That’s just shared features + shared parameters doing their thing. In another word, this is a problem causes by using of NN, but not specific to GRPO.

- This paper use a strong cold start pushes the policy into the “safe regime”, where self-correction already works; this shrinks the headroom for the proposed fix, so gains reflect pretraining quality more than the method. In many deployments you don’t have rich supervised/preference data to seed the policy; if the method shines mainly with good seeds, its robustness from scratch is uncertain.

- The “negative expected advantage” argument is intuitive but glosses over off-policy data, partial observability, and non-stationary judges. I can understand that for theoretical proof, you can somehow assume perfect condition. However, you need to show more experiments to validate the performance across different scenarios, two is not enough.

**Questions:**

- Does your mitigation still help if you replace the policy head with per-action (decoupled) heads or small per-skill adapters?
- A cold start lifts you out of the “danger zone,” so training self-corrects faster and looks more stable. However, this making the proposed fix appear less critical. Based on my understanding, GCD disentanglement should helps the most when starts the danger zone, where gradient coupling dominates and errors amplify; Why you decided to do the experiments with cold starts as this seems not the best way to present the effectiveness of your method?
-I think you need to define the loss of GCD in the main text. L_gcd should be properly defined somewhere.
- is GCD a token level approach as GRPO is at the trajectory level or both is at the trajectory level? In figure 6, it seems that it is all at a trajectory level...which is different with you claim

---

### Official Review · Reviewer_wZax · 2025-10-21

**Soundness:** 1
**Presentation:** 1
**Contribution:** 1
**Rating:** 2
**Confidence:** 4

**Summary:**

The paper investigates the failure of GRPO in agentic RL tasks. It (may) argue that the true cause is not reward miscalibration, but gradient coupling between similar samples. This coupling allegedly causes gradients from good trajectories to incorrectly reinforce similar bad actions. The authors propose an auxiliary classification task, Generative Classification Disentanglement (GCD), to separate the embeddings of good and bad actions and mitigate this issue.

**Strengths:**

While the paper may tackle an important problem, it is difficult to discuss the strengths with the current manuscript now.

**Weaknesses:**

(1) Severe Presentation Issues: The paper is nearly un-reviewable due to its extremely poor writing quality. I feel that the manuscript might be somehow automated or randomly generated. It is filled with severe grammatical errors, non-existent words (e.g., "extrsome", "emely", "simplicify"), and logically broken sentences. For example, a single sentence in the Related Work section is contradictory and incoherent:

>Reinforcement Learning (RL during reinforcement learning) algorithms like PPO (Schulman et al., 2017) is growing extrsome emely popular which would weaken the performancebecause it can greatly help the performance through Reinforcement Learning from Human Feedback (RLHF) (Ouyang et al., 2022a).

Similar issues pervade the entire manuscript. The current version is not above the bar for publication or even review at ICLR.

(2) Incomprehensible Methodology: Because the writing is so unclear, the paper's central claims are difficult to understand, let alone verify. The precise distinction between "gradient coupling" and the "squeezing effect" is not clearly articulated, and the technical implementation of the proposed "Generative Classification Disentanglement (GCD)" is opaque. It is impossible to judge the soundness of the method when it is not described coherently.

**Questions:**

.

---

### Official Review · Reviewer_Y7oa · 2025-10-30

**Soundness:** 3
**Presentation:** 2
**Contribution:** 3
**Rating:** 2
**Confidence:** 3

**Summary:**

The authors want to address a bottleneck in improving the performance of LLMs on agentic tasks: some failure modes increase after GRPO. They show this is not justifiable with reward miscalibration or "squeezing effect" and instead hypothesize that it's because some good actions are similar to bad actions and the positive gradient for the good action makes the bad action more probable too.

**Strengths:**

- An interesting new perspective on LLM training dynamics in a simple model of: probability of the wrong actions and their risk

- Improved empirical performance over baseline GRPO

**Weaknesses:**

- The reward miscalibration hypothesis doesn't assume that we are taking the expectation over all the trajectories. We only cover a part of the trajectory space in a GRPO run and this subspace may reinforce the bad actions that are present in successful trajectories. So the part of the argument that dismisses it is not accurate.

- The learning dynamic in section 3.3 is not mathematically rigorous or backed by empirical data. Providing intuition about the dynamics is great, but it should only be an introduction to the actual proof or evidence.

- I spent more time than needed to understand section 3.3 and the interesting idea behind it; it could be a lot simpler with a set of equations over time $t$. Right now, it's pretty badly written to be honest. Its Figure 4 is also rushed. Also section 4.1 is very vague about $\mathcal L_\text{GCD}$. I've decreased my score by 1 only because of the writing of these two sections.

- Missing ablations about the classification task of GCD. I suspect that an LLM can decide whether an intermediate action is good or bad since it needs planning and rollouts to find out. It's not just LLM judging, say, the quality of a math solution while already knowing the answer.

**Questions:**

- What is the setup for Figure 3b? How are the probabilities in the heatmap measured?

---

### Official Review · Reviewer_AYh5 · 2025-10-30

**Soundness:** 1
**Presentation:** 1
**Contribution:** 1
**Rating:** 0
**Confidence:** 3

**Summary:**

The paper identifies the similarity between prompts across sequences of observations collected by an agentic LLM interacting with an environment to be the main reason which causes bad actions (actions which have a negative effect in the long-term performance) to have positive advantages. Authors relate this issue with subsequent observation-actions pairs having similar gradients although very different consequences in the environment. The authors then propose a method in which the agent LLM also acts as a critic, self-reflecting about the failure modes when interacting in the environment to modify the prompt to more effectively separate observation-action tuples.

Unfortunately, the paper is full of incoherences, syntactical, grammatical and lexical mistakes which render the paper unevaluable. (see Section 2 - paragraph 1 for the clearest example).

**Strengths:**

-

**Weaknesses:**

Unfortunately, the paper is full of incoherences, syntactical, grammatical and lexical mistakes which render the paper unevaluable. (see Section 2 - paragraph 1 for the clearest example).

The method and ideas are not clearly presented. The results presented are not clearly discussed, and conclusions supporting the claims are unclear as well. Additionally, the results do not represent a significant improvement over existing algorithms.

**Questions:**

-

**Details Of Ethics Concerns:**

The paper is written so poorly that it seems likely an LLM was used, and used incorrectly. An example paragraph illustrates this:

"LLM Reinforcement Learning Reinforcement Learning (RL during reinforcement learning) algorithms like PPO (Schulman et al., 2017) is growing extrsome emely popular which would weaken the performancebecause it can greatly help the performance through Reinforcement Learning from Human Feedback (RLHF) (Ouyang et al., 2022a)."

Generally, the paper is full of incoherences, syntactical, grammatical and lexical mistakes which render the paper unevaluable.

---

### Note · Authors · 2025-11-12

**Comment:**

We sincerely thank the reviewers for their valuable feedback. We apologize for any lack of clarity in the original manuscript and will try to improve the presentation and address the concerns in the revised version

**Withdrawal Confirmation:**

I have read and agree with the venue's withdrawal policy on behalf of myself and my co-authors.